# The Interaction of Anti-DNA Antibodies with DNA: Evidence for Unconventional Binding Mechanisms

**DOI:** 10.3390/ijms23095227

**Published:** 2022-05-07

**Authors:** David S. Pisetsky, Angel Garza Reyna, Morgan E. Belina, Diane M. Spencer

**Affiliations:** 1Department of Medicine and Immunology, Division of Rheumatology and Immunology, Duke University Medical Center, Durham, NC 27710, USA; diane.m.spencer@duke.edu; 2Medical Research Service, Veterans Administration Medical Center, Durham, NC 27705, USA; 3Epigenomic Regulation in Development and Cancer Team, Broad Institute of MIT and Harvard, Boston, MA 02142, USA; angel.garza.reyna@duke.edu; 4Duke University School of Medicine, Duke University, Durham, NC 27705, USA; meb126@duke.edu

**Keywords:** DNA, anti-DNA antibodies, systemic lupus erythematosus, avidity, monogamous bivalency

## Abstract

Antibodies to DNA (anti-DNA) are the serological hallmark of systemic lupus erythematosus, a prototypic autoimmune disease. These antibodies bind to conserved sites on single-stranded and double-stranded DNA and display variable region somatic mutations consistent with antigen selection. Nevertheless, the interaction of anti-DNA with DNA has unconventional features. Anti-DNA antibodies bind by a mechanism called monogamous bivalency, in which stable interaction requires contact of both Fab sites with determinants on the same extended DNA molecule; the size of this DNA can be hundreds to thousands of bases, especially in solid phase assays. This binding also requires the presence of the Fc portion of IgG, a binding mechanism known as Fc-dependent monogamous bivalency. As shown by the effects of ionic strength in association and dissociation assays, anti-DNA binding is primarily electrostatic. Like anti-DNA autoantibodies, anti-DNA antibodies that bind specifically to non-conserved sites on bacterial DNA, a type of anti-DNA found in otherwise healthy individuals, also interact by monogamous bivalency. The unconventional features of anti-DNA antibodies may reflect the highly charged and polymeric nature of DNA and the need for molecular rearrangements to facilitate monogamous bivalency; the Fc portion contributes to binding in an as yet unknown way.

## 1. Introduction

Antibodies to DNA (anti-DNA) are the serological hallmark of systemic lupus erythematosus (SLE) and valuable biomarkers for underlying immune disturbances [1,2,3]. A prototypic systemic autoimmune disease, SLE primarily affects young women and is characterized by the production of antibodies to nuclear molecules (antinuclear antibodies or ANAs) [4,5]. These antibodies target proteins, nucleic acids, and complexes of proteins and nucleic acids. While ANA production is almost invariable in patients with SLE, the specificity of these antibodies differs among patients, suggesting the existence of serologically defined disease subsets [6].

In the context of autoimmunity, IgG antibodies are considered to be the most relevant isotype for pathogenicity. While IgM anti-DNA antibodies can also be found in the sera of patients with SLE, these antibodies appear part of a spectrum of antibodies called natural autoantibodies that are a feature of normal immunity [7,8]. These antibodies can bind both foreign and self antigens, albeit at low affinity, and may represent a protective element in the initial host response to infection, pending the induction of high affinity IgG antibodies. While the assay of both IgG and IgM anti-DNA may have utility for assessing disease activity and prognosis [9,10], most available assays detect only IgG antibodies because of their clear association with key events in SLE.

As a group, IgG anti-DNA antibodies bind determinants on single-stranded (ss) as well as double-stranded (ds) DNA, although antibodies to dsDNA are more specific for SLE [1,11]. These determinants, or epitopes, are present on the phosphodiester backbone and are called “conserved,” since they occur on all DNA molecules independent of species origin. In this conceptualization, B-DNA, the classic Watson–Crick dsDNA structure, is the main structural feature recognized by anti-DNA antibodies. Since all natural DNA display the B-DNA conformation, anti-DNA assays have used mammalian, bacterial, and plasmid DNA as antigens [1,2].

Given the prominence of anti-DNA as a marker for SLE, the molecular basis and origin of these antibodies have been the subject of intensive investigation since their discovery almost 70 years ago. These studies, which have involved the analysis of antibodies from both patients and animal models, have led to two major conclusions which are seemingly at variance: (1) DNA is poorly immunogenic; and (2) anti-DNA antibodies arise as a result of antigen selection by DNA. The first conclusion is derived from immunization models which have worked successfully for autoimmune diseases that target protein autoantigens. As demonstrated in these models, DNA, even when presented with a protein carrier and administered with adjuvant, elicits very limited, if any, production of antibodies to B-DNA [1,11].

Despite the poor immunogenicity of B-DNA, anti-DNA antibodies from patients and murine models show evidence of antigen selection as revealed by the molecular properties of antibodies expressed during disease; the most illuminating studies of this kind have characterized monoclonal anti-DNA antibodies derived from inbred lupus mice [12]. These antibodies show an increased content of positively charged amino acids (e.g., arginine) in the third complementary determining region (CDR) of the heavy chain. Furthermore, comparisons of these sequences with those in the germline indicate the role of somatic mutation. As the phosphodiester backbone is negatively charged, an increase in positively charged amino acids is consistent with antigen selection by DNA.

While the role of antigen selection has now become dogma in the field, studies from our laboratory, and those of others, have suggested that the story is more nuanced and complex. DNA–anti-DNA interactions show binding features that are rarely, if ever, encountered with antibodies to other antigens, either self or foreign. In this article, the unconventional binding properties of anti-DNA antibodies will be reviewed to highlight the underappreciated and perhaps unprecedented nature of the anti-DNA response. For convenience, the word serum will be used even if studies utilized plasma; also, some of the studies cited characterized the binding of anti-DNA in serum preparations which are likely collections of different specificities. The term anti-DNA will be used in a general sense. Finally, the seminal contributions of Professor David Stollar are gratefully acknowledged [11]. Professor Stollar is one of the founders of this field and was prescient in using the principles of biochemistry to elucidate the binding properties of anti-DNA antibodies.

## 2. Anti-DNA Antibodies Bind Large DNA

An antibody combining site can accommodate a piece of DNA a few nucleotides (approximately three base pairs) in size. Nevertheless, most anti-DNA antibodies fail to bind to short oligonucleotides in either solid phase or fluid phase assays. In fluid phase assays, a piece of DNA of approximately 35 base pairs in size is required for stable binding, with some anti-DNA requiring pieces of DNA hundreds of bases in size [13]. Such a piece of DNA is much greater in size than a combining site; it is long enough, however, to span the distance between the two antibody combining sites, although this distance varies among isotypes.

The requirement for a large piece of DNA is consistent with a binding mechanism known as monogamous bivalency. In this mechanism, stable binding requires occupancy of both antibody combining sites (or Fab sites) with determinants on the same molecule, in this case, a polynucleotide chain. Because of the low affinity of each combining site, stable interaction requires that both combining sites simultaneously contact determinants along a single piece of DNA [13]. Bivalent binding is a mechanism to increase antibody avidity. While each Fab site may have low affinity, an intact IgG anti-DNA antibody can have high avidity because of the potential for bivalency.

In our studies, we have used solid phase assays to investigate monogamous bivalency, since these assays appear to detect a wider range of avidities than fluid phase assays. Using an ELISA, we analyzed the size requirements of DNA for anti-DNA detection. For antigens, we used *Hinf1* restriction enzyme digests of DNA that had been separated by gel electrophoresis to produce a range of DNA sizes from 100 to 2000 bases. For comparison, undigested DNA had a size of greater than 20,000 bases. For these experiments, we used ssDNA rather than dsDNA and prepared fragments from calf thymus, *E. coli* and salmon testes DNA [14].

The results of these studies were striking: we found greatly reduced antigenicity of DNA preparations below 2000 bases. This size far exceeds the size of antibody combining sites as well as the distance between two Fab sites of a single antibody molecule. Of note, while the fragments were inactive when bound to the solid phase, they were nevertheless able to block the binding of anti-DNA to undigested DNA when used as fluid phase inhibitors.

To explain the large size of DNA needed for binding in an ELISA, we suggested that DNA of a few thousand bases or less becomes tightly adherent to the solid phase over its entire length, leaving very little DNA that can be considered free in the fluid phase or “soluble”. With DNA firmly attached to the solid phase, we proposed that DNA is unable to undergo the structural rearrangements to allow interaction of both Fab sites with a single polynucleotide chain. In contrast, with high molecular weight preparations (as usually used in ELISA for anti-DNA), we suggested that much of the DNA is effectively soluble even if some of the DNA is immobile on the plastic surface.

To explore this possibility, we tested the antigenicity of restriction enzyme digests of biotinylated DNA, attaching it to the solid phase via streptavidin. We found that the biotinylated restriction enzyme digests in the solid phase were effective antigens when bound to streptavidin, even though these digests were inactive when coated to the surface directly. We suggested that, with the attachment via streptavidin, much of the DNA is present in the fluid phase (albeit in close proximity to the plastic surface) and can structurally rearrange to allow monogamous bivalency [14].

To the extent that monogamous bivalency is the basis of anti-DNA interactions, our findings suggest complexity in the way that monogamous bivalency mediates antibody binding to DNA in the solid vs. fluid phase. This difference is important since DNA *in vivo* may be presented on the surface of particles as a kind of corona, or coating, or adherent to the kidney basement membrane as a planted antigen [3,15,16,17]. These findings also suggest that bivalent binding may occur between infrequent or even rare determinants on the phosphodiester backbone, which must be free in the fluid phase to loop or rearrange to bring sites together for monogamous bivalency. Rather than evidence for simple binding to the B-DNA backbone, our findings suggest that anti-DNA interactions may entail binding to variations in backbone structure based on perhaps subtle differences in the spacing of phosphate groups. Given the distance between these determinants, a large antigen may be needed for monogamous bivalency. In this regard, since dsDNA is a rigid molecule, a larger piece of DNA may favor rearrangements by more gradual bending.

While the molecular basis of these interactions is still under investigation, the size requirement for anti-DNA binding contrasts with the binding of antibodies to proteins. While antibodies to proteins likely bind to protein surfaces that depend on conformation, they can also bind to small peptides that represent part of the overall antigenic structure [18,19]. Binding to peptides can nevertheless be sufficiently strong to allow detection under conventional assay conditions, including in the solid phase. Furthermore, immunization with peptides can be used to induce antibodies to proteins, likely because peptides, even if small, can undergo conformational changes to display antigenic determinants similar to those in the full length protein [20].

From these studies, we would conclude that one defining feature of anti-DNA is the requirement for a large or extended antigenic structure for stable binding. It is not clear whether this feature results from fundamental differences in the antigenicity of nucleic acids compared to that of proteins or whether the differences relate to the setting of normal immunity compared to autoimmunity.

## 3. Anti-DNA Antibodies Require the Fc Portion

Monogamous bivalency implies a low affinity of each Fab site for DNA antigen, with simultaneous occupancy of antigenic determinants along the same polynucleotide needed for sufficient avidity for detection in an immunoassay. If that is the case, then an F(ab’)2 fragment of an anti-DNA antibody should bind DNA as effectively as an intact antibody. An F(ab’)2 fragment is bivalent, albeit truncated, compared to an intact antibody. To test the hypothesis that F(ab’)2 antibodies bind DNA, we used venerable biochemical approaches to prepare fragments by digestion of IgG preparations with pepsin; the resulting fragments entirely lack the Fc portion of the antibody. Surprisingly, we found that bivalent F(ab’)2 antibody fragments from IgG from patients with SLE are unable to bind to DNA antigen in an ELISA. Such fragments, however, could bind well to foreign antigens such as tetanus toxoid or EBV antigen preparations; similarly, monovalent Fab fragments, generated by papain digestion, could also bind to the foreign antigens while being unable to bind DNA [21]. These findings point to an idiosyncrasy or peculiarity in antibody interactions with DNA antigen.

For conventional antibodies, the role of the Fc portion of IgG relates to its effector functions such as fixation of complement or binding to Fc receptors of cells. While some studies have indicated that the Fc portion of an IgG can influence the avidity or specificity of antibody binding [22,23,24,25,26], our findings indicate a more absolute requirement of the Fc portion for binding. Models for antibody–antigen interactions do not incorporate a role of the Fc portion for stable binding, raising the questions about its role in anti-DNA binding.

At present, we do not know the basis of the requirement for Fc for anti-DNA binding. We can, however, offer possible explanations. The Fc portion may cause structural changes in the two Fab sites, to increase their affinity or avidity of interaction by an allosteric effect. The second possibility relates to Fc:Fc interactions, with two adjacent antibodies interacting via their Fc portions to increase avidity; this interaction would be comparable to the boost in avidity produced by crosslinking of antibodies by an anti-IgG reagent or rheumatoid factor. Finally, the Fc portion may serve as a third “combining site,” with DNA interacting with Fc along with the two Fab sites in a triangular structure in space. In this case, DNA must be large or flexible enough to touch all three binding elements. Figure 1 illustrates these possibilities.

As noted, while prior studies have indicated that variation in isotype could influence binding properties of an antibody, the constructs studied had an Fc portion whereas our F(ab’)2 fragment was entirely devoid of this antibody component [22,23,24,25,26]. Whatever the mechanism of Fc-dependent monogamous bivalency and the role of Fc in DNA binding, anti-DNA can fix complement, promoting inflammation and damaging tissues. The ability of anti-DNA antibodies to fix complement may suggest that DNA does not bind to Fc strongly in a way that alters the effector function of the antibody including interaction with the complement system.

The expression of antibodies to DNA, while highly associated with SLE, is not unique to this condition. Among other autoimmune diseases, autoimmune hepatitis is notable for the occurrence of anti-DNA responses that can be detected by the same assays as used for determination of anti-DNA responses in SLE [27,28,29,30]. These assays include the *Crithidia lucilae* immunofluorescence assay that is highly specific for antibodies to dsDNA. Nevertheless, patients with autoimmune hepatitis do not develop clinical manifestations similar to those in SLE, most notably, nephritis. It would therefore be of interest to determine whether the anti-DNA antibodies in autoimmune hepatitis and other conditions bind DNA by Fc-dependent monogamous bivalency and can fix complement.

## 4. Antibodies to Bacterial DNA Bind by Monogamous Bivalency and Variably Require Fc

We obtained additional insight into the role of Fc by investigating another type of anti-DNA binding. While antibodies to B-DNA are a unique feature of SLE, antibodies to non-B-DNA structures can occur in otherwise healthy individuals (NHS); these anti-DNA antibodies likely arise as a response to non-conserved determinants on foreign (bacterial or viral) DNA introduced during infection or colonization. As we demonstrated many years ago, the blood of NHS contains high levels of antibodies to DNA from certain bacterial species, including *Micrococcus luteus* (formerly *lysodeikticus*) and *Staphylococcus epidermidis* [31,32]. These NHS anti-DNA antibodies differ from lupus anti-DNA in important respects: they are specific for DNA from certain bacterial species; they do not bind B-DNA; they are predominantly IgG2; and they have a predominance of κ light chain [33]. Importantly, these antibodies are not autoantibodies (even though they bind DNA) and arise in the setting of normal B and T cell function; furthermore, their generation is not influenced or shaped by disturbances in the pre-immune B cell repertoire that may occur in SLE [34]. In at least some patients with SLE, antibodies specific for bacterial DNA coexist with anti-DNA autoantibodies, indicating preservation of a normal mode of B cell recognition of non-B-DNA along with the abnormal B cell recognition of B-DNA.

To determine whether this ”normal” type of anti-DNA interaction also utilizes Fc-dependent monogamous bivalency, we prepared Fab and F(ab’)2 fragments from NHS IgG and tested them for binding to DNA from *Micrococcus luteus* (MC). In our previous studies, we have found that MC DNA is bound by essentially all NHS serum tested and is a prototype of a bacterial DNA antigen. Using an ELISA, we demonstrated that the Fab fragments of NHS IgG failed to bind significantly to MC DNA, resembling in this respect, Fab fragments from SLE IgG in their interaction with B-DNA [35]. These findings indicate that, even with anti-DNA arising in a putatively normal immune system, the affinity of each Fab site remains low, with monogamous bivalency necessary for stable binding. To the extent that antigen selection operates during the generation of anti-DNA to foreign DNA antigen, a conventional high affinity interaction appears difficult to achieve. The fallback position in host defense is monogamous bivalency.

The studies with F(ab’)2 fragments provided a mixed result. The F(ab’)2 fragments of two of five NHS samples tested bound to MC DNA at levels similar to those of the intact IgG. Similarly, two of five of SLE F(ab’)2 samples also bound to MC DNA; for these samples, adsorption of the preparation with calf thymus DNA cellulose was used to eliminate the antibodies to B-DNA that could also bind to MC DNA [35]. Together, these findings suggest that the nature of anti-DNA interactions with B-DNA and non-B-DNA may differ in that an Fc portion is not strictly required for the binding of foreign DNA, pointing to a more conventional mode of interaction.

## 5. Anti-DNA Antibodies Depend on Electrostatic Interactions

The binding of antibodies to antigens depends on the structure, chemical composition and charge of the antigen, with this interaction involving different bond types at places where antibody contacts antigen. While antibody–antigen binding can be analogized to a lock and key, a handshake may also pertain as surfaces of antibody and antigen align [18,19]. The situation with DNA and anti-DNA may be different, however, since DNA is a long and charged molecule and has fewer surface features that would lead to hydrogen bonding or hydrophobic interactions, for example. Furthermore, dsDNA is a rigid molecule, potentially limiting antigen–antibody contacts [1,11].

To delineate further the features of anti-DNA interactions, we explored the effects of ionic strength on the binding of SLE anti-DNA. For this purpose, we varied the salt concentration from 150 mM to 1 M of sodium chloride in a pH 7.5 buffer in the initial binding reaction of the ELISA. As these studies indicated, binding of SLE anti-DNA to DNA is exquisitely sensitive to the salt concentration, with antibody binding diminishing at 250 mM and essentially completely gone with 1 M. These findings indicate a strong predominance of electrostatic interactions in anti-DNA binding, with high salt leading to charge shielding [36].

An additional way to assess the role of electrostatic interactions involves dissociation assays in which the salt concentration is increased following establishment of equilibrium of antibody binding. Using this approach, we showed that, with the exception of one serum of six tested, SLE sera showed a marked loss of antibody binding following the switch to a high salt condition. The salt sensitivity of DNA–anti-DNA interactions was also observed with the NHS anti-DNA response to MC DNA, consistent with electrostatic binding to a charge array, albeit of somewhat different geometry than that of the classic B-DNA [37]. As previously reported for association assays, NHS anti-DNA to MC DNA showed an effect of increased ionic strength in dissociation assays (Figure 2), suggesting a difference in the contribution of electrostatic interactions by the two sources of anti-DNA antibodies.

## 6. Anti-DNA Antibodies Can Show Hysteresis

As noted in the text above, for one of the SLE sera tested, the results of association and dissociation assays differed as this serum showed resistance to the dissociating effects of high ionic strength. This resistance developed in a time-dependent fashion over a period of minutes, with maximum resistance observed after one hour [37]. These findings indicate a slow transition of an electrostatic interaction into another bond type by a mechanism known as hysteresis. Hysteresis, a term usually applied to physical processes such as magnetism, indicates that the state of a system depends on the condition for its development.

While a physical–chemical change of either antibody or antigen could occur with the anti-DNA system, we favor a gradual shift in the conformation of DNA into a non-B-DNA structure as an explanation for our results. The behavior of the serum displaying hysteresis indicates that, while non-electrostatic interactions can develop between DNA and anti-DNA, the initial interactions are nevertheless almost exclusively electrostatic. As sensitivity to salt can reflect the strength of antibody binding, these studies provide further insight into the avidity of anti-DNA binding.

## 7. Conclusions

Anti-DNA antibodies have conventional heavy and light chains; they have somatic mutations indicative of selection and they show features consistent with an interaction with a highly charged molecule. Nevertheless, anti-DNA antibodies bind DNA by mechanisms that can be considered unconventional since they appear so distinct from those of other antibodies ([Table ijms-23-05227-box001]). Indeed, we have not found another example of Fc-dependent monogamous bivalency although, in fairness, few studies have actually looked for this binding mode. Future studies, therefore, are in progress to determine whether the unconventional binding modes of anti-DNA reflect the unique physical–chemical properties of DNA or the influence of the autoimmune state on antibody and specificity.

**Box 1 ijms-23-05227-box001:** Unconventional Features of Anti-DNA Antibodies *.

Fc-dependent monogamous bivalency Requirement for a large antigenic structure for stable binding Lack of binding to small molecule representations of antigen Predominance of electrostatic interactions Hysteresis

* In this context, the term unconventional signifies the distinctive aspects of anti-DNA binding in comparison with those of other antibodies. In some instances, these features have not been investigated for antibodies to foreign antigens or other autoantigens.

## Figures and Tables

**Figure 1 ijms-23-05227-f001:**
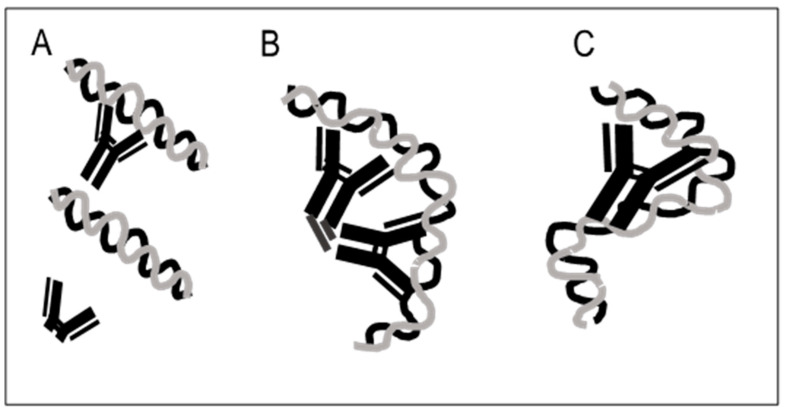
The role of Fc in the binding of anti-DNA antibodies to DNA. This figure illustrates possible mechanisms by which the Fc portion of an IgG anti-DNA can contribute to binding to DNA. Panel (**A**) indicates that the fully intact IgG can bind to DNA while the F(ab’)2 fragment is unable to do so. The failure of the F(ab’)2 to bind could result from an alteration in the Fab binding sites by an allosteric change. Panel (**B**) indicates that Fc-dependent monogamous bivalency results from interaction of both Fc portions, leading to cross-linking of IgG molecules. Panel (**C**) indicates that DNA interacts with both Fab binding sites as well as sites on the Fc portion, effectively creating a third binding site.

**Figure 2 ijms-23-05227-f002:**
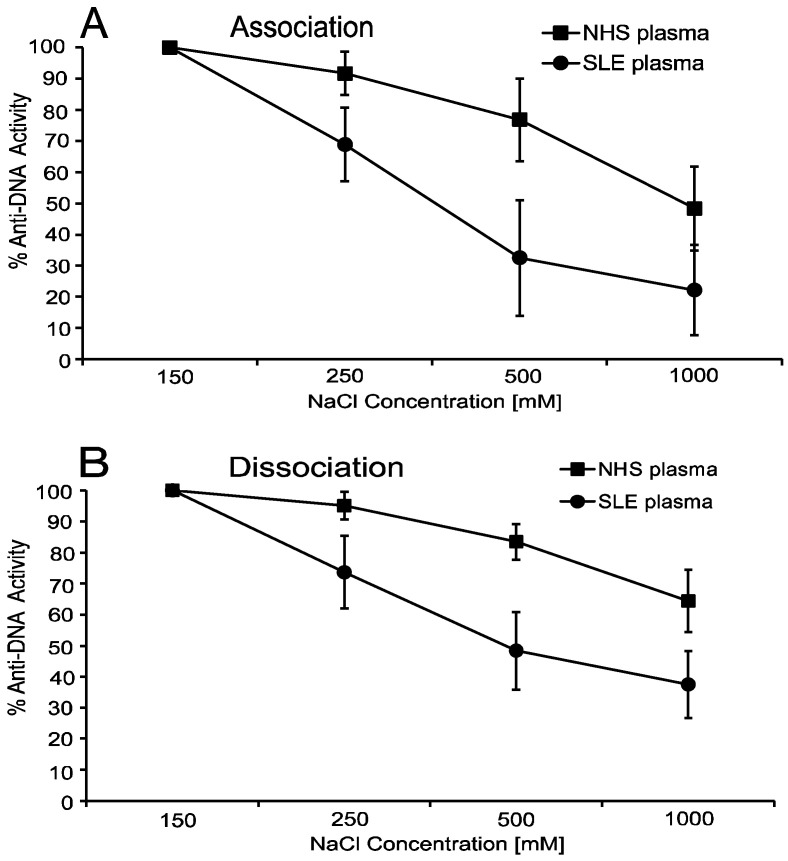
The effects of ionic strength on anti-DNA interactions with MC DNA. The figure illustrates the effects of increasing ionic strength on the binding of NHS anti-DNA and SLE anti-DNA to MC DNA in association (Panel (**A**)) and dissociation (Panel (**B**)) assays. As the data indicate, the binding of NHS anti-DNA was less sensitive to the effects of ionic strength than SLE anti-DNA, suggesting differences in the role of electrostatic interactions in binding. The study involved five SLE sera and four NHS sera. Means and standard deviations are shown.

## Data Availability

The data presented are available on request from the corresponding author.

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
