# Peer review of "The Interaction of Anti-DNA Antibodies with DNA: Evidence for Unconventional Binding Mechanisms"

_ijms, 2022, doi:10.3390/ijms23095227_

Round 1

Reviewer 1 Report

This manuscript by Pisetsky et al. reviews “The Interaction of Anti-DNA Antibodies with DNA: Evidence for Unconventional Binding Mechanisms”. The review article is updated, very well written and includes novel observations made by the authors and other groups. While I have no important criticism to the manuscript, I have a couple of comments/questions from a clinical perspective:

  • Although the presence of anti-dsDNA antibodies is primarily associated with SLE, they can occasionally be found in patients with autoimmune hepatitis, primary Sjögren’s syndrome and antiphospholipid syndrome as well as in subjects exposed to TNF-inhibitors. Can the authors speculative on whether the binding characteristics of anti-dsDNA described here are unique for immunological mechanisms associated with SLE, or if anti-dsDNA antibodies in other diseases may have similar capacities?
  • Are the unconventional binding mechanisms of anti-dsDNA antibodies found in SLE related to their ability to activate complement?
  • In the first part of the text (page 1-2), the authors describe how anti-DNA antibodies are associated with SLE; antibodies against dsDNA much more specific for lupus than those antibodies targeting ssDNA. Although anti-DNA antibodies of IgM class are prevalent in SLE (but less specific), I assume that the authors refer to the IgG isotype. I would suggest mentioning early that they mean the IgG isotype, especially as isotypes are discussed further later in the text.     

Author Response

This reviewer had a number of comments which addressed important issues concerning anti-DNA antibodies.  We appreciate the reviewer noting these issues.  In response to these comments, we have made a number of changes:

We have addressed the issue of isotype in a new paragraph near the beginning of the article.  We discuss the difference between IgM and IgG anti-DNA and the focus on IgG antibodies in terms of pathogenicity.

We have added additional text concerning the relationship between Fc-dependent monogamous bivalency and complement fixation.

We have added a new paragraph on the nature of anti-DNA in other diseases including autoimmune hepatitis.

For each of these new sections, we have provided additional references.

Reviewer 2 Report

The manuscript is well written and complete. 

I recommend its publication in its present form .

Author Response

This reviewer did not have comments to address.